# Live Weight and Sex Effects on Sensory Quality of Rubia de El Molar Autochthonous Ovine Breed Meat

**DOI:** 10.3390/ani11051293

**Published:** 2021-04-30

**Authors:** Eugenio Miguel, Belén Blázquez, Felipe Ruiz de Huidobro

**Affiliations:** Departamento de Investigación Agroalimentaria, Instituto Madrileño de Investigación y Desarrollo Rural Agrario y Alimentario (IMIDRA), Finca el Encín. Apartado 127. Autovía A-2, Km 38200, 28800 Madrid, Spain; blazquezcmb@madrid.es (B.B.); felipe.ruiz@madrid.org (F.R.d.H.)

**Keywords:** meat sensory quality, weight, sex

## Abstract

**Simple Summary:**

Rubia de El Molar is in danger of extinction, and this work is part of a research project funded by the Regional Government of Madrid that aims to know its productive characteristics to prevent its disappearance. The aim of this work is to study the effect of weight and sex on suckling lamb meat sensory characteristics of Rubia de El Molar breed. According to sensory characteristics, two weight groups of animals were detected: light carcasses (10 and 15 kg) and heavy carcasses (20 and 25 kg). Heavy carcasses’ meat was tougher, more elastic, and received lower pleasantness scores than light animals. Sex had a lower effect on meat sensory parameters than weight, and meat from male lambs showed higher flavour intensity and also higher pleasantness scores than female meat.

**Abstract:**

Fifty-six Rubia de El Molar ovine breed lamb carcasses were divided into 8 groups (*n* = 7 per group) according to weight (10, 15, 20 and 25 kg) and sex (male and female) to study the effect of these factors on meat sensory characteristics, assessed by means of a trained panel. Heavier animals showed a more-springy meat and also received lower scores for pleasantness. Assessors gave lower scores for flavour and pleasantness in female meat. A high correlation was detected between hardness and springiness, number of chews, and pleasantness. Juiciness, greasiness, and flavour were also sensory variates highly correlated. Pleasantness was only correlated to hardness. Changes from 10 to 25 kg did not affect juiciness, greasiness, flavour, and number of chews. There are no significant differences in the sensory quality of lambs slaughtered at 10 and 15 kg live weight, and also between 20 and 25 kg groups. Heavy carcasses (20 and 25 kg) showed a more hard and springy meat than light carcasses (10 and 15 kg). Besides, the 10 and 15 kg animals group received higher scores for pleasantness. This work showed differences in suckling lambs’ meat sensory parameters between Rubia de El Molar and other ovine breeds.

## 1. Introduction

In Madrid, there are several autochthonous breeds: a goat breed (Cabra del Guadarrama goat) and two ovine breeds (Rubia de El Molar and Colmenareña). Both autochthonous ovine breeds are phylogenetically related to the Churra breed. Rubia de El Molar is listed as very endangered because it has a census lower than 1000 animals throughout the Madrid Region (Rubia de el Molar is one of the less numerous ovine breed groups in Spain). Rubia de El Molar is a dual-purpose sheep: milk and meat.

Carcass weight is an important cue in the categorisation of lamb meat. In the Mediterranean countries, like Spain, lambs are slaughtered at very young ages, producing light carcasses. These animals are usually fed with concentrate feed in addition to their mother’s milk. In Spain, the market demands light carcasses, and meat from light lambs is considered better than meat from heavier animals. Four-hundred consumers were asked to evaluate different attributes (price, certification, origin, and commercial type) of lamb meat. Results obtained by means of conjoint analysis techniques show that regular consumers as well as occasional ones show a preference for lamb meat type. In this sense, a market share simulation of preferred (suckling and “ternasco”) types proved that regular consumers generally prefer suckling lamb to “ternasco” lamb [1].

Several previous works of our group showed that weight and sex had an effect on the prediction of tissue composition in suckling lamb carcasses using the European Union scale [2,3,4]. The commercial value and consumer acceptance of lamb meat are mainly determined by carcass weight and sex, which can predict tissue distribution in the carcass and sensory characteristics of meat and eating quality. Lamb meat quality depends on sex, breed, and weight, among other factors [5,6,7,8].

To describe meat characteristics, different analyses are needed: chemical, physical, and sensorial [9]. Characteristics directly related to the physical components of meat products can be measured through instrumental methods. In addition to these physical components, sensory parameters (including texture, flavour, and appearance cues) can be better described by human subjects. To test sensory parameters, a sensory analysis by trained panellists is the most appropriate tool. Sensory panels are essential because of providing complementary information to instrumental methods [10].

Rubia de El Molar meat quality was also studied by instrumental methods by our group [11]. Both instrumental and sensory studies contribute to obtain specific and important information on overall meat quality, and a meat sample might be described with different techniques, resulting in many diverse characteristics. A relationship might exist between instrumental measurements and sensory panel evaluations because each type of analysis studies the same properties by means of different methods.

## 2. Materials and Methods

Fifty-six Rubia de El Molar sucking lambs were used in this work. All the animals came from an experimental herd of Rubia de El Molar ovine breed, conserved in “El Encín” farm belonging to the Madrid Institute of Agricultural and Food Research (IMIDRA). The animals were reared in an intensive regime. The treatments studied were sex (males and females) and slaughter live weight (10, 15, 20, and 25 kg of live weight). Animals were milk-fed until slaughter at a commercial slaughterhouse. During this period, concentrate feed was supplied to the lambs ad libitum in addition to their mothers’ milk. The composition of the concentrate feed changed during lamb development: a starter feed was supplied during the first weeks and a growth feed until slaughter. Chemical composition of the concentrate feed is given in Table 1. Most of the lambs selected for this study came from single birth, except for 9 of them who were twins but were raised as single lambs. Animals were slaughtered in a commercial slaughterhouse, in accordance with the rules of Spanish legislation regarding the transport and slaughter of slaughter animals [12]. Seven males and seven females were included in each slaughter weight group.

### 2.1. Carcasses

Carcasses were held at 4 °C for 24 h and then were properly transported to IMIDRA.

### 2.2. Sampling and Sensory Analysis

Samples from m. Longissimus thoracis et lumborum muscle (L1–L6) were used for sensory analysis. These samples were vacuum packaged and deep-frozen (−40 °C) until analysis. Meat was thawed in cold water for 2 h, and then meat samples were packed and grill-cooked as follows: samples were packed on aluminium-folded strips and grill-cooked to a core temperature of 75 °C in a preheated grill (300 °C), as a modification of the method described in [13] to the method described in [14]. Samples were stored in a preheated oven at 75 °C until tasting. No salt was added. Sensory analysis was performed either on 1 × 1 × 1 cm cubes (for all sensory parameters except the number of chews), or on cylinders of 10 mm height and a diameter of 15 mm (for the number of chews).

Sensory quality of meat was assessed by means of a sensory panel of trained assessors. Panel members were previously selected and trained for meat sensory analysis. Different foods were used as standard to build sensory characteristics scales for hardness, springiness, juiciness, greasiness, and flavour [13]. The use of reference scales produces more closely grouped assessor’s scores [2,15], and these scales have proven to be useful when assessing sensory characteristics [13,16].

Unstructured scales 10 cm long were used to assess intensity of every parameter. However, previous training had provided assessors some standard food, representing intensity points ranging from 1 to 5. Assessors pointed out each parameter having in mind the levels of intensity of food parameters, and they made a trait on the scale, being free to use every scale point, where the distance to the left extreme of the scale was then measured by means of a ruler.

Although some authors recommend not asking hedonic questions to trained assessors [10], because it is a fact that this kind of judgement does not match consumer responses to food, in descriptive-quantitative analysis, it is a common practice to make a question on the pleasure/displeasure they experience during meat consumption. This is a parameter that informs us about the global acceptance by well-trained people of the meat consumed during a sensory analysis.

### 2.3. Statistical Analysis

Data were statistically analysed by means of the statistical package Statistica for Windows, r. 5.0 [17,18]. Statistical analysis included principal components analysis. Sensory characteristics’ data were analysed by means of ANOVA and correlation analysis. The effect of sex and weight on sensory quality of the Rubia de El Molar lamb meat was studied. An analysis of variance was carried out considering sex and weight as fixed effects. The interaction of both factors was also studied. In the case of the effect of weight, after the analysis of variance, a Tukey test for multiple comparisons was performed. For sensorial analysis results, session effect and assessor effect have been tested by means of a fixed ANOVA performed on the scores of each assessor in each session. The ANOVA was performed taking all effects as fixed effects because assessors were not randomly chosen assessors were just highly trained people, and they worked as a team as a measure instrument.

The factor analysis was based on the principal components analysis (PCA) by means of Statistica for Windows, r. 5.0 multivariate plot (principal component analysis option). PCA tested all sensory variables included in the work, and also sensory variables and zootechnical numerical variates (carcass weight).

Correlation analyses were performed between assessors’ scores for the sensory variates and also with zootechnical numerical variates (carcass weight).

## 3. Results

### Sensory Analysis

Results obtained for sensory analysis of meat are shown in Table 2. Live weight had a great effect on hardness and springiness. Hardness scores were greater as live weight increased. Heavier animals that showed a more-springy meat also received lower scores for pleasantness. Scores for hardness were lower (*p* ≤ 0.001) for meat of lambs slaughtered at 10 and 15 kg than for meat of lambs slaughtered at 20 and 25 kg. Statistically significant differences for springiness were only observed between 10 and 20 kg groups. Animals slaughtered at 10 kg showed lower scores (3.25 points) than those slaughtered at 20 kg (4.23 points).

Scores for pleasantness were always lower for meat of animals slaughtered at a lower live weight (*p* ≤ 0.01).

Meat from male and female were only different for pleasantness and flavour. Assessors gave lower scores for flavour (*p* ≤ 0.05) and pleasantness (*p* ≤ 0.01) for female meat. Meat from male lambs received 6.04 points for pleasantness and 5.52 for flavour, while meat from female lambs received 5.46 points for pleasantness and 5.06 for flavour. We found no differences for hardness, springiness, juiciness, greasiness, and number of chews between female and male meat. So, sex had no effect on meat sensory hardness or meat sensory springiness. An effect of sex is only shown on pleasantness and flavour. We found no statistically significant differences in juiciness, greasiness, flavour, and number of chews (assessed by a trained panel) amongst live weight groups.

In order to study the relationship between sensory variates in Rubia de El Molar suckling lambs’ meat, a principal components analysis (PCA) and a correlation study were performed. The results of the PCA for Rubia de el Molar suckling lambs’ meat sensory parameters are presented in Table 3.

The first principal component (PC1) was softness, and it is characterised by hardness and springiness. A high coefficient is observed for greasiness. This first PC explains 36.83% of the variation observed. The second PC (PC2) is defined by juiciness. A high coefficient is detected for greasiness and number of chews. Greasiness and flavour are equally important for both PCs, and pleasantness has a higher relation with tenderness than with juiciness. This second principal component explains 27.5% of total variation observed. The first two PCs explain 64% of the variance observed.

Figure 1 shows plots of the measurements of Rubia de El Molar suckling lambs’ meat quality variates on the first two PCs. Measurements and PCs are interpreted according to the correlations between parameters. Measurements close to each other are positively correlated, measurements separated by 90° are independent, and measurements separated by 180° are negatively correlated. Greasiness, juiciness, and flavour are placed in the loading plot on the top (indicating their importance for the second PCs). These three variates are allocated in a different place than springiness, hardness, and number of chews, that are also grouped on the bottom of the loading plot. These three variates show a negative correlation to greasiness, flavour, and number of chews. Pleasantness is placed in a different allocation on the right of the loading plot. Pleasantness shows a negative correlation to springiness, hardness, and number of chews, and a positive correlation to greasiness, juiciness, and flavour. Pleasantness has a greater importance when defining the first PC than the second PC.

The results of the PC analysis for carcass and sensory parameters in Rubia de El Molar suckling lambs’ meat are presented in Table 4. Including weight in the principal components analysis, three factors were detected. The first principal component was characterised by live weight. High coefficients were observed for hardness and pleasantness. This first principal component explained almost 40% of the total variation observed.

The second PC was defined by greasiness and flavour, and a high coefficient was observed for juiciness. This second principal component explained 18% of the total variation observed. The third PC is characterised by number of chews. The number of chews is the number of molar strokes necessary to reduce food to a state that makes it fit for swallowing. It is the only sensory parameter assessed that is not evaluated by using reference scales to assess the intensity of parameters, and some food as standards for scale points. High coefficients were observed for hardness and springiness. A higher number of chews is required when meat is hard and springy, and also when it is less juicy. This third PC explained 15% of the total variance observed.

Figure 2 shows plots of the measurements of Rubia de El Molar suckling lambs’ zootechnical and sensory meat quality variates on the first two PCs. Greasiness, juiciness, and flavour are also placed in the loading plot on the top, defining the second PCs. Weight variates are allocated on the right in the loading plot, defining the first PC. Pleasantness has greater importance defining the first PC than the second PC.

Figure 3 displays the projection of the sensory quality data in the first two PCs. Ten- and fifteen-kilogram animals tended to be situated on the right side of the plot (all animals but two, and all animals but three for 10 and 15 kg respectively, are there). However, 20 and 25 kg groups tended to be situated on the left side of the plot. An effect of sex is observed for the group of animals of 10 kg, and females and males are placed in different locations in the plot. As for the two-dimensional graph, PC1 and PC2, we note that there was considerable scattering in the first component for female, while the observations for male are more clustered.

Figure 4 displays the projection of the sensory quality data in the first two PCs for male and female groups. No separated groups can be observed, although males tended to be located in the central area of the plot between females.

According to our results, a trained taste panel can discriminate animals of different weight for the Rubia de El Molar ovine breed. However, sex differentiation is less clear.

A statistically significant positive correlation is observed between carcass weight and sensory parameters hardness and juiciness, and a negative correlation to pleasantness (Table 5). Hardness scores were greater as live weight increased. Heavier animals showed a more-springy meat that also received lower scores for pleasantness.

A median correlation is detected between hardness and springiness (r = 0.56), hardness and number of chews (r = 0.54), and hardness and pleasantness (r = −0.54). Meat that is perceived as tougher by the judges is also perceived as more-springy, and both attributes cause a greater number of chews required before swallowing. The meat that had greatest scores for hardness and springiness received lower scores for pleasantness by the assessors. No statistically significant correlation is observed between hardness and juiciness or greasiness and flavour. Pleasantness is only correlated to hardness. Juiciness, greasiness, and flavour are highly correlated sensory variates. As greasiness increases, meat is perceived as juicy and higher scores for flavour are observed. These three sensory parameters were not correlated to pleasantness.

The most important factor for pleasantness for Rubia de El Molar suckling lambs’ meat was, however, hardness. Only a statistically significant correlation between pleasantness and hardness was observed. The other sensory parameters (springiness, greasiness, juiciness, flavour, and number of chews) were not correlated to pleasantness. Besides, the results of the PC analysis for Rubia de El Molar suckling lambs’ meat sensory parameters show that the first principal component is characterised by hardness and springiness. A high coefficient is observed for greasiness. The second PC is defined by juiciness. A high coefficient is detected for greasiness and number of chews. Greasiness and flavour are equally important for both PCs, but pleasantness has a higher relation with tenderness than with juiciness.

There are no significant differences in the sensory quality of lambs slaughtered at 10 and 15 kg live weight. Similarly, there are also no differences between the groups of animals of 20 and 25 kg. We have grouped the lambs of 10 and 15 kg in a class of light lambs and 20 and 25 kg in another class of heavy lambs. Table 6 shows the results of comparing the sensory qualities of these two groups of lambs. Animals slaughtered at a weight lower than 15 kg showed a softer meat, less springy, that received higher scores for juiciness and pleasantness than the meat of lambs slaughtered at 20 and 25 kg. According to our results, it is recommended to choose 10–15 kg as slaughter live weight for Rubia de El Molar lambs to obtain a better-quality meat.

## 4. Discussion

Live weight had a great effect on hardness and springiness. Hardness scores were greater as live weight increased. Heavier animals that showed a more-springy meat also received lower scores for pleasantness. No differences were detected for greasiness, flavour, or number of chews between weight groups. Yousefi et al. showed that meat from moderate carcass weight (15–20 kg) had higher overall acceptability scores compared with light (10–15 kg) and heavier carcass weight (20–25 kg) in fat-tailed Chall lambs [19]. Lower scores for tenderness were obtained for meat of medium-weight lambs (30–31 kg) when compared with low-weight (25–26 kg) and high-weight (35–36 kg) groups in an assay, with fifty-two male Kivircik lambs, set up to investigate the influences of weight and production system (concentrate-based system, or pasture-based system) on carcass and meat quality characteristics [20].

These results obtained for Rubia de El Molar ovine breed meat differ from those obtained for Rasa Aragonesa lambs [21]. Sensory traits were not influenced by carcass weight (with the exception of juiciness). There were no differences due to carcass weight in organoleptic scores assigned for tenderness, flavour, and overall satisfaction. In this work, three carcass weight groups were formed: group A with an average carcass weight of 8.07 kg, group B (10.22 kg), and group C (13.42 kg) [21].

Sensory parameters were affected by age at slaughter in Altamurana lambs. Results of the sensory analysis showed that meat from lambs slaughtered at 40 days was more tender and chewable, but less juicy and fatty than meat from lambs slaughtered at 75 days. No significant effect of age at slaughter was perceived by the trained panel on flavour, odour, sour, sweet, and bitter parameters [22].

Tejeda et al. [23] did not observe significant differences between classes of hot carcass weight in lamb juiciness, but light animals exhibited slightly higher values than did heavy animals, which we also observed in our study (2.73, 2.82, 3.33, and 3.38 points for 10, 15, 20, and 25 kg groups, respectively).

Other authors did find differences due to the weight of the animals in the sensory quality of lamb meat. So, lamb and fat odour, and metallic and acid flavour intensities, were greater in the heavier carcasses in other work, also in Rasa Aragonesa lambs, although these changes did not affect overall acceptability [24]. Meat from suckling lambs (11.4 kg) of two breeds, Grazalema-Merino (dairy breed) and Churra-Lebrijana (meat breed), obtained higher scores for sustained juiciness and were more tender than light lambs (20.5 kg) [25]).

We only observed a sex effect on sensory parameters of Rubia de el Molar suckling lamb meat for flavour and pleasantness. Assessors gave lower scores for flavour and pleasantness for female meat. The results of published studies on the effect of sex on the sensory qualities of lamb meat generally agree that differences between sexes in sensorial meat quality are generally not important [23,26].

Dransfield et al. [27], Sañudo et al. [5], and Teixeira et al. [8] did not observe significant sensory differences between meat from male and female lambs. As we showed for the Rubia de El Molar ovine breed, several studies suggest that the effect of gender on the sensory profile of lamb’s meat is relatively small. Statistically significant differences have been demonstrated in some studies [28,29,30]. Flavour intensity seems to be the quality trait most influenced by sex. Channon et al. [31] reported that flavour intensity was greater in males than in females, with differences being more evident in adults than in young animals. Flavour intensity is also the sensory attribute most often influenced by sex [29]. The differences in the sensory profile between genders seem to be more apparent in older than in younger lambs. Meat from male lambs (52.2 kg) had higher scores for cloying and rancid flavour, and lower scores for sour and sweet taste compared to female lambs’ meat (47.9 kg) using Norwegian White lambs [32]. Meat from female lambs has a higher intensity in sweetness and sourness, and a lower intensity in cloying flavour than meat from male lambs. No differences in sensory parameters (aroma, succulence, flavour, hardness, and overall acceptance) were obtained between male and female groups in an assay to study to characterise the meat quality and the global acceptance of three Brazilian native ovine breeds (Morada Nova, Rabo Largo, and Santa Inês) that were managed in semi-intensive systems and their crosses [33].

The eating quality of lamb meat of three indigenous ovine breeds was studied by Arsenos et al. [29]. These authors showed an effect of slaughter weight on sensory characteristics, with decreasing scores for acceptability in older/heavier lambs. Degree of maturity affected flavour, tenderness, and overall acceptability; however, as we also observed for the Rubia de El Molar ovine breed, a sex effect was only detected for flavour.

Effects of sex and carcass weight on the sensory attributes of Terrincho lamb meat, a product with Protected Designation of Origin (PDO) labelling, were studied [34]. No sex effect on meat quality was detected, but the panel could differentiate that meat of heavier animals was toughest and had more intense odour and flavour, whilst light animals were most succulent. These authors also observed that the most important factor for overall acceptability was flavour [34].

The sex effect on sensory parameters for meat of indigenous Pantaneiro sheep and Texel or Santa Inês crossbred finished on feedlot was low [35]. No differences were found for WSF, juiciness, texture, and palatability. Only the meat of females was tender. We found no statistically significant differences in juiciness, greasiness, flavour, and number of chews amongst live weight groups.

The results obtained for sensory texture parameters are correlated to those observed for instrumental texture properties or Rubia de el Molar suckling lamb meat [11]. These results of sensory hardness are correlated with those obtained for these same animals in instrumental texture analysis. Slaughter live weight had a significant effect on Warner-Bratzler shear force in raw meat. Differences arose between 10 and 20 kg groups. Higher values were obtained for 20 kg animals. No effect of weight on the results obtained after performing the Braztler-Warner test on cooked meat was observed. An effect of weight on TPA parameters was only observed for hardness performed on cooked meat [11].

We found no statistically significant differences in juiciness, greasiness, flavour, and number of chews (assessed by a trained panel) amongst live weight groups. Also, no effect of slaughter live weight on moisture content could be observed [11]. Water-holding capacity expressed as a percentage of juice expelled under pressure was affected by slaughter live weight, noting that 20 and 25 kg lambs had a higher proportion of liquid expelled (15.39% and 17.15%, respectively) and therefore were less able to retain water (*p* ≤ 0.001) than 10 and 15 kg lambs (11%, 52%, and 13.15%, respectively) [11]. Also, slaughter live weight influenced cooking losses so that 15 kg lambs showed the greatest losses (*p* ≤ 0.01), differentiating these animals from the rest [11].

An effect of sex was only shown on pleasantness and flavour. We also showed that sex had no effect on moisture content, water-holding capacity, or cooking losses [11].

We had not measured intramuscular fat content (IMF) in longissimus dorsi muscle, but we did for samples obtained from the quadriceps femoris, biceps femoris, semimembranosus, and supraespinatus, as a function of slaughter weight and sex [11]. The percentage of intramuscular fat was influenced by slaughter live weight in the case of the biceps femoris (*p* ≤ 0.001) and supraespinatus (*p* ≤ 0.001), resulting in a significant increase of IMF at 15 and 20 kg (from 2.31% to 2.97% at 15 and 20 kg respectively, in the case of the biceps femoris, and from 2.61% to 3.49% respectively, in supraespinatus muscle) (*p* ≤ 0.001). Differences in tenderness observed as live weight increased could be associated to differences in intramuscular fat content.

A statistically significant positive correlation was observed between carcass weight and sensory parameters hardness and juiciness, and a negative correlation to pleasantness for Rubia de el Molar suckling lamb meat. Hardness scores were greater as live weight increased. Heavier animals showed a more-springy meat that also received lower scores for pleasantness. A positive relationship between tenderness and live weight was detected in [36]. These two parameters were also positively related with the amount of fat [37]. No significant differences in tenderness in samples with differences in fatness were observed by Sañudo et al. [38]. Also, we observed a positive correlation between weight and hardness and juiciness, but no correlation between hardness and juiciness or between hardness and greasiness.

We found no statistically significant differences in flavour amongst live weight groups. However, Teixeira et al. [8] showed that heavy carcasses had more flavour intensity that the light ones studying the effect of weight, sex, and breed on eating quality of Mirandesa and Bragançana lambs. No sex effect on eating quality was observed.

A high weight can be associated with greater flavour intensity [38]. This effect has also been reported in [39,40]. However, Sañudo et al. [5] and Tejeda et al. [23] did not find significant effects of age and weight on the flavour intensities of lamb meat. Muela et al. [24] showed that low hot carcass weight tended to have better overall acceptability, although without statistically significant differences between the two hot carcass weight classes (either ≤10.5 or ≥12.0 kg), but lower tenderness scores, which suggests that tenderness is important, but is not the only factor that influences the acceptability of lamb meat. The importance of factors other than sensory texture on acceptability is particularly true in lamb, in which flavour can be considered the most important [38], determining that, in general, consumers prefer lamb with low odour and flavour intensities [23,38,41,42], although clusters of consumers can be found with different preferences related to their social and their previous experiences with meat [41].

Some authors think that palatability or eating quality includes three main components: texture, juiciness, and flavour/odour characteristics, at least in red and poultry meats. Ekiz et al. [43] found statistically significant correlations between flavour intensity and flavour quality and overall acceptability. Similar results were obtained in [41]. We reported that pleasantness is only significantly correlated to hardness.

Consumers in Anglo-Saxon countries are accustomed to an intense flavour in lamb meat, whereas this is normally rejected in Mediterranean countries [44]. Martínez-Cerezo et al. [45] obtained similar results, with lighter lambs (20–22 kg) being more highly scored, as did Sañudo et al. [46] in comparing lambs of 10–12 kg live weight and 13–15 kg live weight. Tejeda et al. [23] studied the effect of live weight and sex on physico-chemical and sensorial characteristics of Merino lamb meat. Meat quality was not significantly affected by slaughter weight or sex, although meat from lighter lambs (24 kg) had greater general acceptability than meat from heavier lambs (29 kg). Also, meat from the heavier lambs had poorer scores than meat from the lighter lambs [23].

As previously reported, assessors gave higher scores for pleasantness to male lambs’ meat. A positive correlation was also shown between carcass weight and the sensory parameters hardness and juiciness, and a negative correlation to pleasantness. Hardness scores were greater as live weight increased. Heavier animals showed a more-springy meat that also received lower scores for pleasantness.

The results presented in this work suggest different sensory characteristics of the Rubia de El Molar suckling lambs autochthonous ovine breed. In this sense, the most direct ancestor of the Rubia de El Molar breed could be the blonde-faced Latxa ovine breed, with which it bears a lot of character similarity. The differences observed in meat sensory parameters between Rubia de El Molar and other ovine breeds may be due to its smaller size and to the fact that it is a breed with double aptitude: dairy and meat, and also because it is an indigenous breed, different to breeds that have been improved to increase milk or meat production. In Madrid, Rubia de el Molar lambs are slaughtered with a mean carcass weight of 5.5 kg. The main use of the Latxa breed is the production of milk for the production of ewe’s cheese.

The weight of Rubia de El Molar males is 70 kg and that of females 45 kg. [47]. Therefore, at the same weight at slaughter, females have reached a higher percentage of the weight of the adult format. The weight of the adults is less than that of other breeds widely distributed in Madrid and Spain. Thus, Manchego lamb has an adult weight of 100 kg for males and 75 kg for females [47]. At the same weight, lambs of ovine breed with smaller size frame are older, so they have deposited a greater amount of fat and this can affect the toughness, juiciness, greasiness, springiness, and pleasantness of meat. Besides, typically, selected breeds for meat production have a greater number of muscle fibres and a smaller amount of intramuscular fat per muscle unit area [48]. To study the effect of sex and weight on sensory characteristics of meat of different breeds is difficult because the results could be dependent on some criterion, such as same carcass weight, same age, same degree of maturity, etc. [48].

## 5. Conclusions

Slaughter live weight had a greater effect on Rubia de El Molar sensory meat quality than sex. Heavier animals showed a harder and springier meat. Assessors gave lower scores for pleasantness to heavier animals. Assessors also gave lower scores for flavour and pleasantness for female meat. A statistically significant correlation was observed between hardness and springiness, hardness and number of chews, and hardness and pleasantness. No statistically significant correlation was observed between hardness and juiciness, hardness and greasiness, or hardness and flavour. In addition, juiciness, greasiness, and flavour showed a statistically significant correlation. Pleasantness was only correlated to hardness.

A taste panel can discriminate between different weight animals. There were no significant differences in the sensory quality of lambs slaughtered at 10 and 15 kg live weight. Similarly, there were also no differences between the groups of animals of 20 and 25 kg. We grouped the lambs of 10 and 15 kg in a class of light lambs and 20 and 25 kg in another class of heavy lambs

Heavy carcasses (20 and 25 kg) showed a more hard and springy meat than light carcasses (10 and 15 kg). Besides, the 10 and 15 kg animals group received higher scores for pleasantness than the rest.

According to our results, it is recommended to choose 10–15 kg as slaughter live weight for Rubia de El Molar lambs to obtain a better sensory quality in meat.

Rubia de El Molar is a small-size, not improved, dual-purpose (dairy/meat) ovine breed. These characteristics could explain the differences observed between sensory parameters for Rubia de El Molar and other breeds.

## Figures and Tables

**Figure 1 animals-11-01293-f001:**
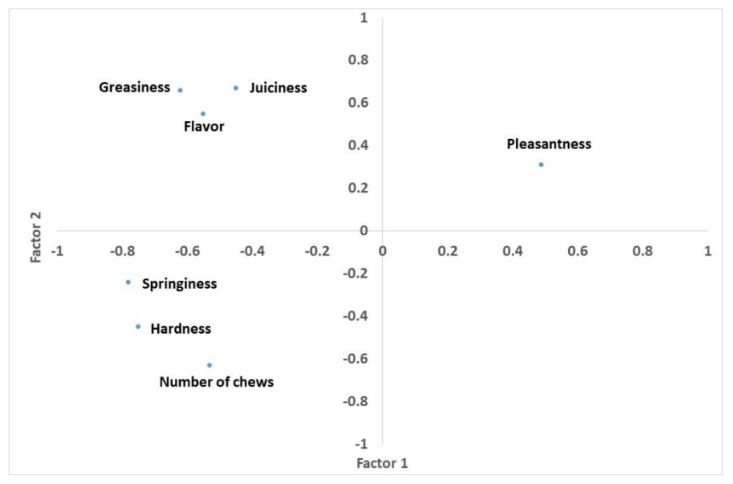
Projection of the sensory quality measurements in the plane defined by the first two principal components.

**Figure 2 animals-11-01293-f002:**
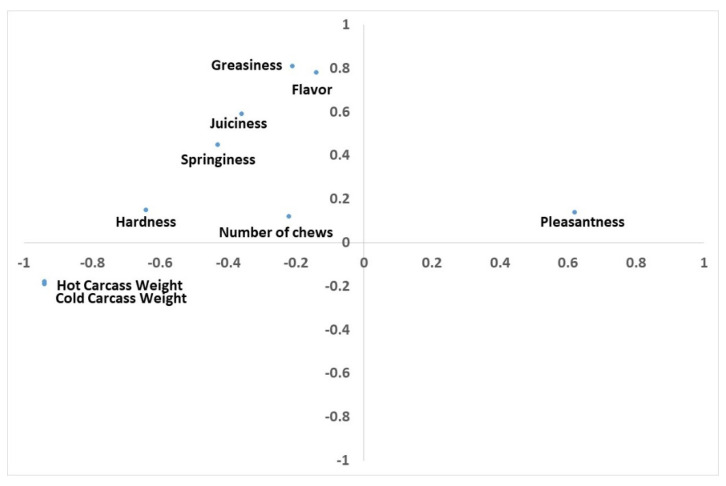
Projection of the sensory and zootechnical quality measurements in the plane defined by the first two principal components.

**Figure 3 animals-11-01293-f003:**
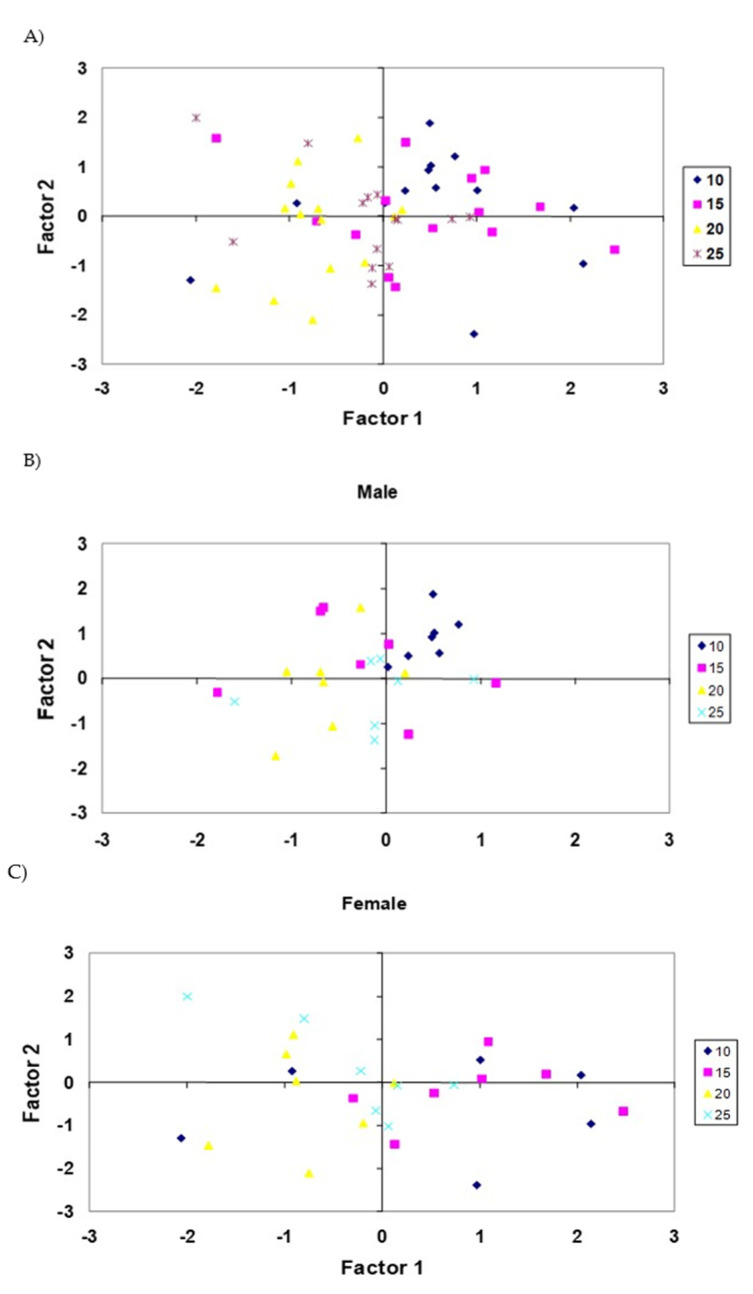
Projection of the sensory quality data of the 10, 15, 20, and 25 kg groups studied in the plane defined by the two principal components. (**A**) All the animals studied in this work, (**B**) male group, (**C**) female group.

**Figure 4 animals-11-01293-f004:**
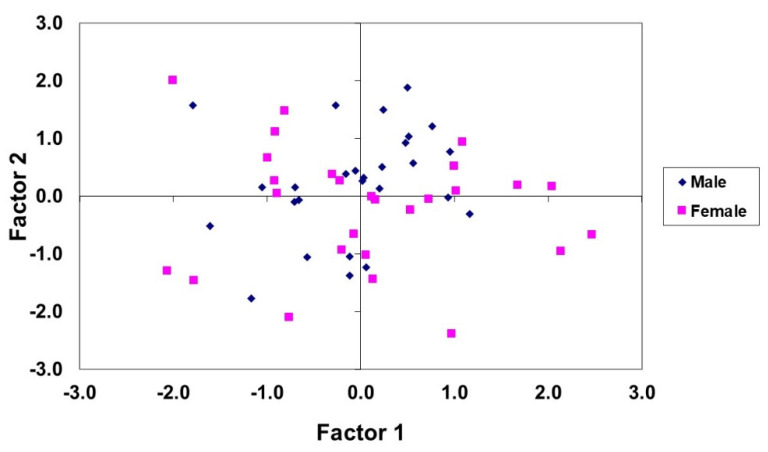
Projection of the sensory quality data of the male and female groups studied in the plane defined by the two principal components.

**Table 1 animals-11-01293-t001:** Chemical composition of the concentrate feed.

Components of the Concentrate Feed	Initiation Feed	Growth Feed
Crude protein (%)	19.00	18.00
Crude fat (%)	5.90	6.00
Crude cellulose (%)	3.50	4.50
Crude ash (%)	7.00	7.00
Starch (%)	31.09	31.09
α-tocopherol (mg/kg)	29.00	29.00
D_3_ Vitamin (IU/kg)	2000.00	2000.00
A Vitamin (IU/kg)	10,000.00	10,000.00
Calcium (%)	0.90	0.90
Phosphorus (%)	0.40	0.40
Natrium (ppm/kg)	1500.00	2700.00
Copper (mg/kg)	5.00	0.00

The data are expressed as a percentage of the weight of feed or mass of the components per kg of feed.

**Table 2 animals-11-01293-t002:** Means and mean square error (MSE) of the sensory parameters of Rubia de El Molar lambs’ meat in the four weight groups and in male and female groups.

Sensory Parameters	Liveweight				Sex		Sig.			
	10 kg, *n* = 13	15 kg, *n* = 14	20 kg, *n* = 14	25 kg, *n* = 14	Male, *n* = 28	Female, *n* = 27	Weight	Sex	W × S	MSE
Hardness	3.27 ^a^	3.47 ^a^	4.63 ^b^	4.23 ^b^	3.86	3.94	***	NS	NS	0.76
Springiness	3.25 ^a^	3.51 ^ab^	4.23 ^b^	3.67 ^ab^	3.78	3.55	*	NS	NS	0.77
Juiciness	2.73	2.82	3.33	3.38	3.22	2.91	NS	NS	NS	0.71
Greasiness	2.96	2.71	3.05	2.97	3.04	2.80	NS	NS	NS	0.81
Pleasantness	6.35 ^a^	5.94 ^ab^	5.38 ^b^	5.33 ^b^	6.04 ^a^	5.46 ^b^	**	**	NS	0.61
Flavour	5.29	5.13	5.48	5.25	5.52 ^a^	5.06 ^b^	NS	*	**	0.57
Number of chews	21.29	18.30	23.00	20.49	20.86	20.67	NS	NS	NS	22.72

Sig. = Level of significance; *** (*p* ≤ 0.001); ** (*p* ≤ 0.01); * (*p* ≤ 0.05); NS: non-significant. MSE = mean square error. M: male; F: female. W × S: Weight × Sex interaction. Means in the same row with different letters are significantly different (*p* ≤ 0.05).

**Table 3 animals-11-01293-t003:** PCA (principal components analysis) results for the first two principal components of sensory parameters in Rubia de El Molar suckling lambs’ meat.

	Factors	
Variates	1	2
Hardness	−0.75	−0.45
Springiness	−0.78	−0.24
Juiciness	−0.45	0.67
Greasiness	−0.62	0.66
Pleasantness	0.49	0.31
Flavour	−0.55	0.55
Number of chews	−0.53	−0.63
% of total variance	36.83	27.53
Cumulative % of total variance	36.83	64.36

**Table 4 animals-11-01293-t004:** PCA results for the first four principal components of zootechnical and sensory parameters in Rubia de El Molar suckling lambs’ meat.

Variates	Factor		
	1	2	3
Slaughter liveweight	−0.94	−0.18	0.20
Weight	−0.94	−0.18	0.19
Hot carcass weight	−0.94	−0.19	0.20
Cold carcass weigh	−0.94	−0.19	0.21
Hardness	−0.64	0.15	−0.58
Springiness	−0.43	0.45	−0.57
Juiciness	−0.36	0.59	0.45
Greasiness	−0.21	0.81	0.20
Pleasantness	0.62	0.14	0.24
Flavour	−0.14	0.78	0.11
Number of chews	−0.22	0.12	−0.85
% of total variance	39.78	18.50	15.45
% of total cumulative variance	39.78	58.29	73.74

**Table 5 animals-11-01293-t005:** Correlation coefficients between zootechnical and sensory variates for Rubia de El Molar lambs’ meat.

	SW	HCW	CCW	Hardness	Springiness	Juiciness	Greasiness	Pleasantness	Flavour	Number of Chews
SW		0.96 *	0.96 *	0.44 *	0.23	0.31 *	0.05	−0.46 *	0.00	0.06
HCW				0.44 *	0.22	0.27 *	0.07	−0.50 *	0.03	0.05
CCW				0.44 *	0.21	0.27 *	0.07	−0.50 *	0.02	0.04
Hardness					0.56 *	0.04	0.18	−0.52 *	0.15	0.54 *
Springiness						0.23	0.26	−0.23	0.25	0.57 *
Juiciness							0.62 *	−0.02	0.34 *	−0.16
Greasiness								−0.13	0.63 *	−0.05
Pleasantness									−0.02	−0.18
Flavour										0.00

SW: slaughter live weight; HCW: hot carcass weight; CCW: cold carcass weight. * Statistically significant correlation (*p* ≤ 0.05).

**Table 6 animals-11-01293-t006:** Means and mean square error (MSE) of the sensory parameters of Rubia de El Molar lambs’ meat in the light and heavy classes of lambs.

Sensory Variates	Live Weight				Sig.	MSE
	Light 10 + 15 kg, *n* = 27		Heavy 20 + 25 kg, *n* = 28		Weight	
	Mean	SD	Mean	SD		
Hardness	3.36 ^a^	1.03	4.43 ^b^	0.73	***	0.79
Springiness	3.38 ^a^	0.93	3.95 ^b^	0.83	*	0.78
Juiciness	2.79 ^a^	0.83	3.35 ^b^	0.89	*	0.74
Greasiness	2.84	1.00	3.01	0.78	NS	0.80
Pleasantness	6.15 ^a^	0.90	5.35 ^b^	0.72	***	0.66
Flavour	5.24	1.02	5.37	0.65	NS	0.73
Number of chews	19.67	5.34	21.74	4.41	NS	23.90

Sig. = significance; *** (*p* ≤ 0.001); * (*p* ≤ 0.05); NS: non-significant. MSE = mean square error. means in the same row with different letters are significantly different (*p* ≤ 0.05). ^a,b^ means in the same row with different letters are significantly different (*p* ≤ 0.05).

## Data Availability

Data sharing not applicable.

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
