# Peer review of "Live Weight and Sex Effects on Sensory Quality of Rubia de El Molar Autochthonous Ovine Breed Meat"

_animals, 2021, doi:10.3390/ani11051293_

Round 1

Reviewer 1 Report

Comments appears on the file annex

Author Response

Thank you very much for the work of reviewing the manuscript. Your comments have been of great help to improve it. We have tried to answer your questions and improve the background and form defects of the manuscript.

We answer your arguments in an attached word file.

Best regards

Reviewer 2 Report

I reviewed your manuscript. I consider that the rescue of indigenous breeds is important for many reasons: cultural, ecological, management, etc. However, it causes me conflict that 56 lambs from a breed of sheep in danger of extinction were slaughtered. How is this fact justified?

In the other hand, other aspects related to the manuscript are the following:

  • Place the figures and tables immediately after they are mentioned in the text.
  • Delete paragraphs of one, two or three lines; it would be better to make one only with the related ideas.
  • Lines 44-45: The last sentence is out of place.
  • Lines 58-64: Citations are missing.
  • Line 77: I think that 56 is the correct number of lambs.
  • Lines 77-92: Some phrases are repeated (lines 81-82 y 89-90, and 83 y 89).
  • Lines 86-87: concentrate’s ingredients are missing.
  • Table 1: what are crude cellulose and crude ash?
  • Line 96: The first sentence was mentioned in lines 90-92
  • Line 120: Cite the authors.
  • Table 2: Superscripts are missing in pleasantness and flavor.
  • Line 250: Table 5 is missing.
  • Table 6 is not mentioned in the manuscript.
  • Methodology and laboratory analysis are not mentioned for intramuscular fat content.
  • References: numbers are repeated and DOI is missing.

Author Response

(The authors gave the same response as above.)

Round 2

Reviewer 2 Report

Dear authors, I appreciate your reply.

Regards.

Author Response

Thank you very much for your comments. I am attaching a file with the answers to your questions.
